

# FLMatchQA: a recursive neural network-based question answering with customized federated learning model

Saranya M and Amutha B

Department of Computing Technologies, School of Computing, SRM Institute of Science and Technology, Kattankulathur, Chennai, Tamilnadu, India

## ABSTRACT

More sophisticated data access is possible with artificial intelligence (AI) techniques such as question answering (QA), but regulations and privacy concerns have limited their use. Federated learning (FL) deals with these problems, and QA is a viable substitute for AI. The utilization of hierarchical FL systems is examined in this research, along with an ideal method for developing client-specific adapters. The User Modified Hierarchical Federated Learning Model (UMHFLM) selects local models for users' tasks. The article suggests employing recurrent neural network (RNN) as a neural network (NN) technique for learning automatically and categorizing questions based on natural language into the appropriate templates. Together, local and global models are developed, with the worldwide model influencing local models, which are, in turn, combined for personalization. The method is applied in natural language processing pipelines for phrase matching employing template exact match, segmentation, and answer type detection. The (SQuAD-2.0), a DL-based QA method for acquiring knowledge of complicated SPARQL test questions and their accompanying SPARQL queries across the DBpedia dataset, was used to train and assess the model. The SQuAD2.0 datasets evaluate the model, which identifies 38 distinct templates. Considering the top two most likely templates, the RNN model achieves template classification accuracy of 92.8% and 61.8% on the SQuAD2.0 and QALD-7 datasets. A study on data scarcity among participants found that FL Match outperformed BERT significantly. A MAP margin of 2.60% exists between BERT and FL Match at a 100% data ratio and an MRR margin of 7.23% at a 20% data ratio.

## INTRODUCTION

Question answering (QA) aims to provide appropriate answers to user-posed NL questions. QA is a widely used and essential artificial intelligence (AI) technique that has garnered significant attention in the academic and industry communities in recent years because of its enormous prospective advantages to practical applications like Google Assistant, Apple's Siri, Amazon Alexa, and other assistants providing information with high intelligence (*Chen et al., 2019*). QA searches articles for relevant responses to a particular topic. It is a field of study that intersects two well-known research areas: natural language processing

Corresponding author
Saranya M, sm2317@srmist.edu.in

(NLP) and information retrieval (IR). In contrast to search engine-performed document retrieval, question answering is based on extracting pertinent, concise replies that are more topic-specific than long, topic-related materials (*Yang et al., 2019*). With rare exceptions, supervised learning problems involving labelled data are commonly associated with quality assurance (QA). Federated learning (FL) (*Ng, Teo & Kwan, 2000*), a unique machine learning technique that preserves anonymity, has recently attracted much attention from researchers. Most machine learning models are trained in a single location, giving the model owner complete access to all training data. On the other hand, FL uses a decentralized method for training models. When clients engage in the most widely used FL approach, they get access to a global model through a central orchestrating server. The most popular FL technique is using a central orchestrating server to give participating clients access to a global model. The local data from these consumers is then used to train the models. After receiving the updated local model parameters, the central server uses them to compile and update the global model by combining the model parameters from each client. This cooperative learning technique protects privacy in three main steps: The current global model G is made available to all participating users by (i) a centralized server, also called a server agent (SA); (ii) the users use their local data to train the received model G, and (iii) they upload their locally trained models Gi back to the centralized server so that they can be combined and updated to create a new global model. Thus, this method is viable for cutting FL's communication costs.

Existing AI models in QA methods face numerous restrictions, mainly stemming from federal data dependency and privacy concerns. Traditional QA techniques need access to huge centralized datasets, which may not constantly be possible owing to data privacy rules or the inaccessibility of varied datasets. Furthermore, centralized techniques face tasks in familiarizing with the degrees of local contexts and languages. FL develops as a promising alternative, leveraging decentralized training through manifold devices while maintaining data privacy. FL permits methods to be proficient collaboratively on local data, modifying the essential for central data sources and addressing privacy concerns. By harnessing the cooperative knowledge of varied datasets across dispersed devices, FL provides a possible solution to the restrictions of centralized QA methods, fostering flexibility, scalability, and privacy protection.

This work presents a unique hypernetwork-based FL framework, the User Modified Hierarchical Federated Learning Model (UMHFLM), that uses the information of various data distributions on clients to get beyond the restrictions that have been identified. Instead of simply fitting a single global adapter to all heterogeneous data distributions, the essential idea is to build the adapter parameters suited to each client *via* hyper networks by taking the information of client data distribution (Fig. 1).

The rest of the article is organized as follows. The first section covers the introduction of this research study. The second section provides the related works and the proposed model is discussed in third section. Then, fourth section gives the result analysis and final section concludes the article.

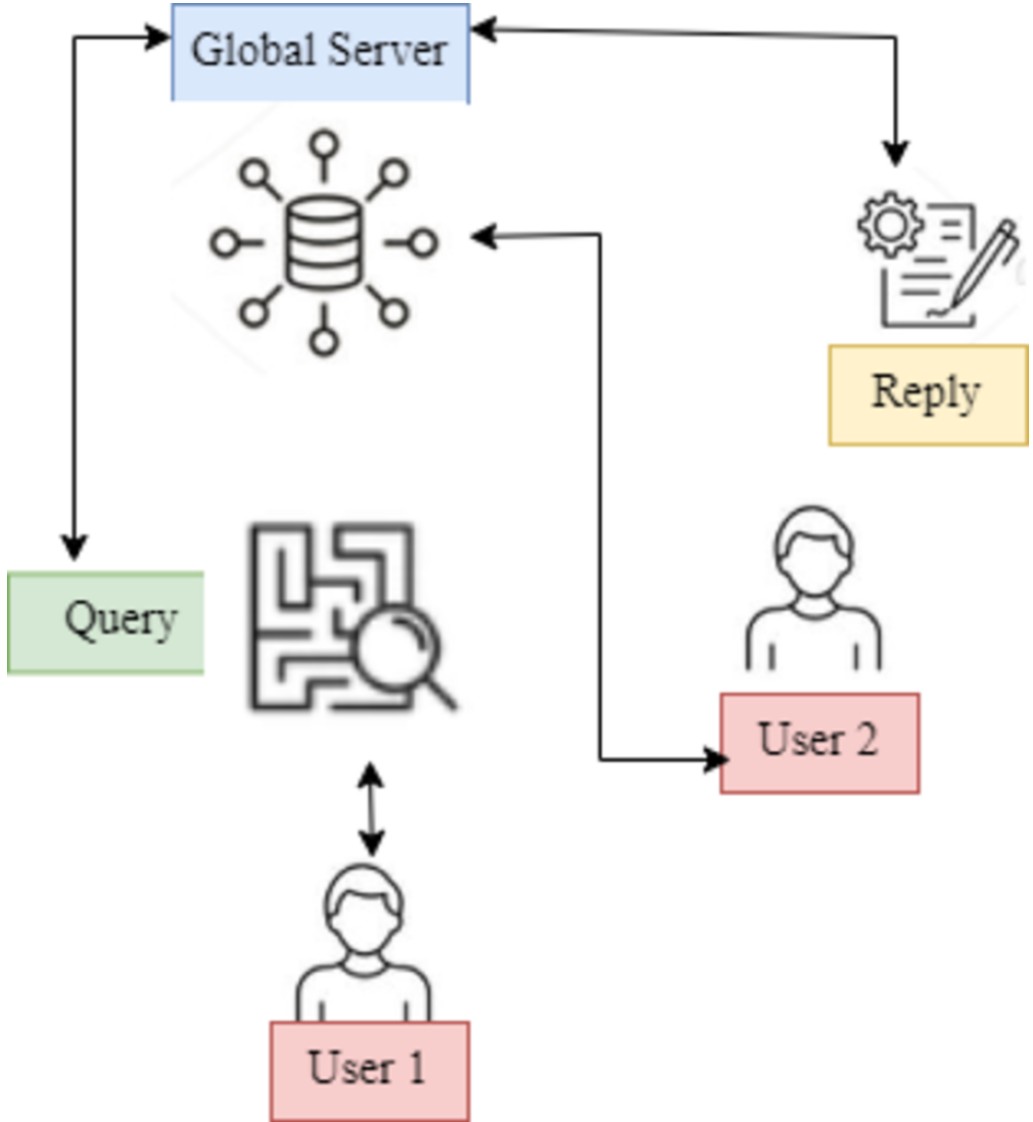

**Figure 1** **UMHFLM's conceptual illustration.** Data distributions on clients to overcome restrictions. Instead of fitting a single global adapter to all heterogeneous data distributions, hyper networks are used to build adapter parameters for each client based on client data distribution information.

## LITERATURE SURVEY

Federated learning (FL) is a privacy-preserving machine learning (ML) method where a central server coordinates a loose federation of local clients. It aggregates local model updates to train a global model while each client keeps its local dataset. FL allows sharing without data leaks. However, traditional FL can leak training data unexpectedly. Differential privacy or robust aggregation can be used for federated model privacy and integrity. In user-level privacy, median aggregation can replace average (*Bonawitz et al., 2021*). A single client stores data for ML model training, which the model owner can view.

Collecting diverse and rich data sets to train robust DL models is difficult. This issue was addressed by *McMahan et al.*'s (*2016*) decentralized learning model. FL lets any number of clients train ML models with local data and share model updates during training. Raw data exposure is reduced. FL's challenges include unbalanced non-IID client data, high client participation, and increased communication costs in the federation (*Zhang et al., 2016*). Recent articles suggest class imbalanced learning in non-FL settings involves separating representation and classification phases, creating high-quality classification representations during representation learning, and balancing decision boundaries between dominant and minority classes during classification, according to FL research (*Narayanan, Rao & Prasad, 2021*). *Elgohary, Zhao & Boyd-Graber (2018)* uses FL settings to select clients with complementary class distributions for updates and requires them to reveal their distribution to the server to address class imbalance using an auxiliary dataset. A hierarchical attention network for answering questions in narrative paragraph reading comprehension is introduced (*Rajpurkar et al., 2016*)—horizontal and vertical attention and fusion across layers at different granularities. SemBERT (*Wang, Yan & Wu, 2018*), a contextual semantics-aware BERT backbone, is also introduced in a survey of 31 QA systems (*Wang et al., 2017*) compared database interfaces, open domains, ontologies, and web document focus. They also detailed the systems and their success rate and corrected answers.QA aims to answer user questions correctly using feature engineering in early research. Statistical syntax-based models softly align questions with answers, while WorldNet's lexical semantic information improves matching (*Kim et al., 2023*). A machine translation model converts query and answer terms, introducing synonyms (*Kim, 2014*). Feature-based approaches are laborious and challenging in capturing semantic information between questions and answers. Recent studies like BERT and RoBERT have introduced (*Lewis & Mensink, 2012*) a value-shared weighting and question term importance-based attention-based neural matching model (*Li, Yu & Dai, 2023*; *Kacupaj et al., 2020*) using large-scale labeled data without considering distributed and isolated data issues. The author proposes adding a proximal term to local objectives to address non-IID data and heterogeneous updates. They use server and client control to estimate update directions, mimicking centralized methods and normalizing and scaling client updates before updating the global model (*Usbeck et al., 2023*). *Azad & Deepak (2019)* surveyed Natural Language Interfaces for databases (NLIDB) for QA systems, not KGQA systems, using a set of 10 questions to evaluate 24 QA systems and compare them to other SQL query conversion systems. ML tasks performed well on the SQuAD dataset, released less than a year ago. A logistic regression (LR) model based on linguistic features by *Liu et al. (2023)* in June 2016 achieved an F1-score of 51%, up from 20%. The author reaches 77.3% F1 using BiDAF encoding, a bidirectional LSTM, and multistage decoding (*Chen et al., 2017*). Microsoft Research Asia's top model, submitted five days before this article, scored 84% F1 on the SQuAD dataset, closer to human reading comprehension at 91% F1 (*Ren et al., 2022*). *Chen et al. (2021)* present to accept FL for QA with the singular concern on the arithmetical heterogeneity of the QA data. Here the heterogeneity mentions the fact that marked QA data are classically with non-identical and independent distribution (non-IID) and unbalanced dimensions in practice. *Lin et al. (2021)* provide the FedNLP, a benchmarking structure

for assessing FL models on four dissimilar task inventions such as sequence tagging, text classification, seq2seq, and question answering. *Chen et al. (2024)* intend a fine-tuning structure personalized to varied multi-modal FL, named Federated Dual-Adapter Teacher (FedDAT). Particularly, this technique influences a Dual-Adapter Teacher (DAT) to find out the data heterogeneity by legalizing the consumer local upgrades and using mutual knowledge distillation (MKD) for an effective knowledge transfer. *Shamsian et al. (2021)* developed a new technique for this issue utilizing hyper networks, called pFedHN for modified federated hyper networks. In this model, a central hyper network system is proficient in producing a set of methods, one approach for every client. In *Qu et al. (2022)*, the authors reveal that self-attention-based architectures (*e.g.*, Transformers) are stronger in delivery shifts and hence recover FL over assorted data. Concretely, the authors conduct the primary rigorous experimental study of dissimilar neural structures through an assortment of federated techniques, real benchmarks, and assorted data splits.

## PROPOSED METHODOLOGY

This section presents the User Client Modified Hierarchical Federated Learning Model (UMHFLM) over the heterogeneous QA data.

### Task definition

Quality assurance focuses on assessing a user's response to a question for relevancy. Define $Task_{QA}$ members in this article. They each have a private QA dataset SQuAD2.0; $D_{task} \in \{D_{task}\,1,\ldots D_{task}N\}$. Equation (1) describes the training dataset $D_{task\_}Training$ from participant ' $t$ ':

$$D_{task}^{Training} = Query_{task}^n, Answer_{task}^n, Reply_{task\,n-1}^{n\,D_{task}^{Training}} \tag{1}$$

where, $Query_{task}^n, Answer_{task}^n, Reply_{task}^n \in [0,1]$ stand for the $i^{th}$ question, user answer, and matching score $D_{task}^{Training}$ samples, respectively. The annotated QA data in real-world scenarios usually have non-IID and unbalanced sizes (*Abebe Fenta, 2023*). They are typically delicate and private in the interim. Consequently, without disclosing the original data, the objective is to create a trustworthy, individual QA model for every participant that incorporates all participant information.

### Overview of the model

This work presents a new federated learning matching system for QA, *"FLMatch,"* to address the FL for QA in heterogeneous settings by using dispersed QA datasets in a privacy-preserving manner to quantify the relevance between questions and responses. In particular, this study considers the quality assurance model for every participant, which comprises both shared and private modules (*Crane, 2018*). This allows participants to efficiently utilize the knowledge of other participants while still capturing the unique features of the local data. Thus, the fundamental concept that drives the design of the *F*LMatch framework is illustrated in Fig. 2.

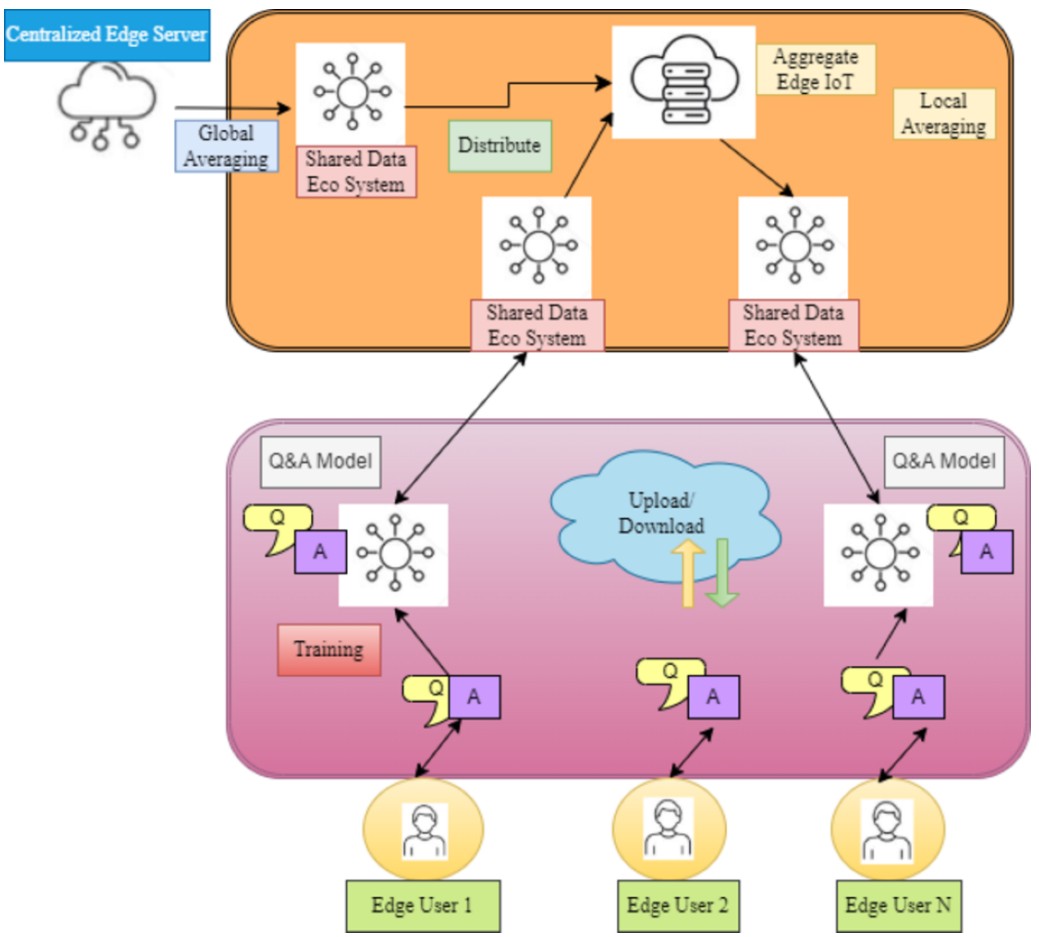

**Figure 2 A comprehensive FL model that aligns with data flow.** Federated learning matching system for QA, "FL Match," to address FL in heterogeneous settings by quantifying question-response relevance using dispersed QA datasets confidentially.

### Common information features

A single participant's labeled data typically needs to be revised to train a reliable quality assurance model. In the *F*LMatch structure, to gather the shareable QA matching expertise amongst many participants, a shared backbone is proposed to mitigate the data sparsity issue. This model uses a cutting-edge neural matching model to determine if a potential response to a question is relevant.

### Domain knowledge maintenance

Sharing the same model amongst participants may not be the best course of action because the QA data held by each participant may differ in size and properties. To mitigate the statistical heterogeneity, this article utilizes a private patch tailored to the unique domain information of each participant. Each participant's backbone is patched, and the patch is trained only using the relevant private local QA data. As a result, the features particular

to each participant are evaluated by the patch component and help to create a distinctive model for them.

### Privacy-based learning

By sharing all model parameters or training samples, participants may compensate for a lack of data, even at the expense of their privacy. Thus, this model suggests employing FL (*Shahamiri, 2021*) technology to maximize *F*LMatch's performance. This work merely uploads the local shared module's parameters—typically containing less sensitive data about privacy—to the central server.

## Federated learning

When training ML models, data are typically stored at a single client, allowing the model owner unrestricted access to the data. Since sensitive data are often involved, gathering rich and diverse data sets can be challenging. This makes it difficult to train effective DL models requiring extensive and varied data sets. FL decentralizes ML model training to solve this issue. Many users can participate in FL's ML model training process. Every client uses its local data to train a model and then distributes the model updates (like model parameters) to other users. After that, additional places can be used, and the model changes can be made. One of FL's primary features is that clients only share model updates rather than local raw data. By doing this, the chance of raw data being exposed is reduced. Many issues can arise in a federated setting, including imbalanced non-IID client data, many clients involved, and expensive communication expenses. Furthermore, the model formalizes FL by rewriting the objective in Eq. (2). This model assumes that there are 'K' users, with 'k' users holding $P_k$ of data. Each user calculates the average loss on user 'k', or $F_k(w)$. $n_k$ or $n_k = |P_k|$, is the number of training samples for each user. 'n' represents the total number of samples for all K users. In FLAvg, a central orchestrating server is required. Motivated by prior research utilizing decentralized training algorithms (*Ida, 2012*), this study expands FLAvg to function in an End-to-End scenario, eliminating the requirement for a central server. FLAvgP2P is another name for the algorithm that was developed. Every client in FLAvgP2P has a unique model and interacts with other users directly. Every user model starts with the same weight, $w_0$, before training. Each user **U** in a round 'x' trains the model using its local data $L_d$, producing a model. Next, every user aggregates and compiles updates from randomly selected neighbors. Where **C** is the fraction of neighbors (*e.g.*, 0.1) and **N** is the total number of neighbors in the network for that client. $R_t$ where $|R_t|$ is computed by $f_n$. After that, the local model is modified. The total number of neighbors in the network is represented by **N** of that client, and **C** is the proportion of neighbors (*e.g.*, 0.1). The local model is then updated,

$$w_{x+1}^{f_n} = \frac{N_R W_x^k}{N_X} + \sum_{k-R} \frac{n_k}{n_x} w_x^k \tag{2}$$

where $N_k$ represents the number of samples at neighbor 'k', and $n_x$ is the total number of representatives from user **U** and the clients in $R_x$. **X** is the local minibatch size; **Y** is the number of times each client trains over the local data set per round, *i.e.*, epoch; and '$\eta$'

is the learning rate. FLAvgP2P shares four hyperparameters with FLAvg. In Algorithm-I, FLAvgP2 is present.

*Algorithm: Federated averaging peer-to-peer (FLAvgP2P)*

Step 1. User 'U' executes:

Step 2. **Initialize** $W_r = 0$

Step 3. **For Each** round $x = 1, 2, \ldots$ **Do**

*Step 4.* $X = $ (split $P_R$ into batches of size B)

Step 5. **For Each** local epoch, i from 1 to E, **Do**

step 6. **For Each** batch, **Do**

Step 7. $W_x^R = W_x^R - \eta \Delta l(W_x^R; Y)$

Step 8. $M = Max\{R, X, 1\}$

Step.9. $R_x = $ (Random Set of R neighbors)

Step10. **For Each** User $k = R_x$ **Do**

Step.11. $W_x^R = AssignWeight(k)$

Step.12. $w_{x+1}^{f_n} = \frac{N_R W_x^k}{N_X} + \sum_{k-R} \frac{n_k}{n_x} w_x^k$ Step.13. **End**

In this work, several participants' backbone-patch architecture-based QA models are trained on data *via* FL to protect their privacy. A central server in *FLMatch* manages several clients for backbone sharing and patch updates, as seen in Fig. 3. In this case, the clients are distinct QA participants who use privately held data to train their models. The following actions are part of the training phase, which begins with the server randomly initializing the shared backbone's parameters '$p$':

1.  To train the models for the following round, the server assigns the global shared backbone's parameters, or '$p$', to each client.

2.  By adding a private patch to the shared backbone, each client uses privately stored data to train their local models. Let '$px$' formally represent the local shared backbone parameters for each client '$x$', and $\beta_x$ represents the private patch parameters. The loss function for every client '$x$' is defined by Eq. (3), a pairwise ranking loss across the training dataset.

$$D_{task}^{Training} \tag{3}$$

where a QA matching model is indicated by $f$ Regarding the question $Q_x$, the symbols $a+x$ and $a-x$ indicate a pertinent response and a negative response, respectively. All the parameters of the local client model are shown by $p_x = (p_x, \beta_x)$. To be more precise, '$p$' and '$\beta_x$' are initialized randomly while $\theta$ $t$ is initialized using the parameters of the standard model, Eq. (4).

$$LengthQ_x(Qx, A_x^+, A_x^-, P_x) = Max(Q1 - f(Qx, A_x^+) + f(Qx, A_x^-) \tag{4}$$

(3) Minimizing a global loss overall '$d$' distributed clients is the objective for each training period (*i.e.*,), Eq. (5)

$$Min\{P_1 \ldots .p_d\} Length(P_1 \ldots .P_r). \tag{5}$$

After every training period, every client notifies the server of the updated local shared backbone's parameters $p_x$.

(4) After gathering parameters from every client, the server updates the global backbone while monitoring each client for parameter aggregation. Formally, this work changes

**Peer**J Computer Science

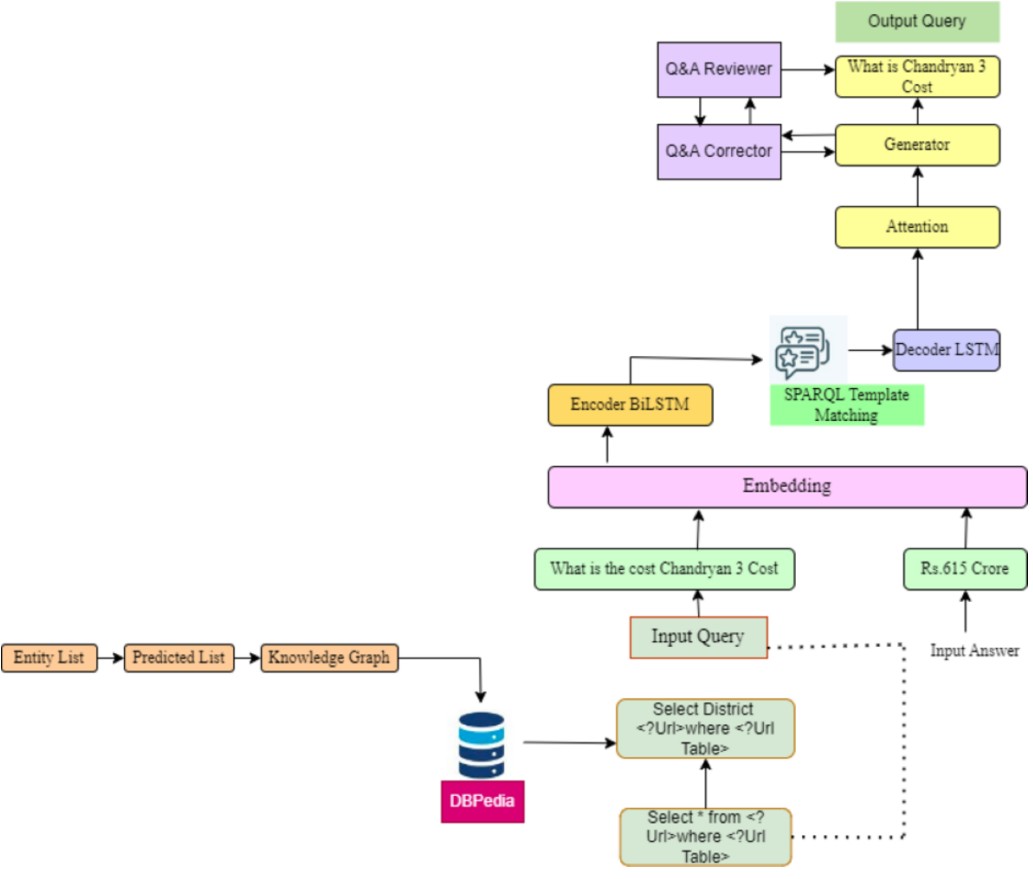

**Figure 3 SQuAD2.0 dataset generation workflow.** In order to train their models, the clients are distinct participants in quality assurance who use data that is privately held.

the globally shared backbone's parameters that are kept on the central server based on the information provided by the **U** clients, Eq. (6)

$$R = \frac{1}{N} \sum_{N-1}^{R} P_x.$$

(6)

## UMHFLM

This model goes into detail on the proposed framework in this section. To lessen the detrimental effects of client heterogeneity, the fundamental technique is to create individualized UMHFLM modules for each customer. First, this work builds latent vectors that reflect each client's data distribution to accomplish this. The settings of the UMHFLM modules are then customized for each client by conditioning the resultant embeddings on the hypernetworks (*Zhang et al., 2023*). This article successfully factorizes the hypernetworks' weights, considering the numerous parameters that are derived from them. This model contains the formation of modified UMHFLM units personalized to

every user's exact necessities. Originally, latent vectors are built to arrest the shades of every customer's data distribution, permitting modified method selection for users' tasks. Next, the UMHFLM modules' formations are modified for every customer by conditioning the resultant embedding on hypersystems. This procedure emphasizes a complete approach to find customer heterogeneity within FL structures, offering comprehensive insights into both the ranked FL construction and the growth of client-specific adapters, thereby improving flexibility and performance across assorted consumer bases.

### Adapter architecture

Determining the structure of the UMHFLM modules that will be developed is the first step in this process. Although several modules have been presented, this study concentrates on the Adapter because of its versatility in various domains, including audio and vision-and-image, and its proven effectiveness in completing specific duties. Within each block of the UMHFLM, the adaptor comprises feed-forward and down-projection functions interspersed between the feed-forward and self-attention layers. The process of adapting can be expressed as shown in Eq. (7)

$$Adption^l(x) = U^l GeLU\left(d_x^l\right) + x \tag{7}$$

where, $d_x^l$ represents the weights for the down- and up-projection in the UMHFLM's $l^{th}$ layer, respectively; $d$ denotes the UMHFLM's hidden dimension and 'b' the bottleneck dimension.

### User embeddings' construction

This work considered two forms of information to represent the clients' characteristics: (a) label embeddings and (b) context embeddings (*Duan et al., 2021a*; *Duan et al., 2021b*). The explicit information on class distribution for each client is communicated in part through label embedding. The label distributions on mini-batches can adequately represent the data distributions of clients because they are often sampled using a uniform distribution. As a result, this model builds label embeddings using the mini batches' label distributions. The label embeddings can be obtained in Eq. (8) if Batch = Di represents the client i's mini-batches:

$$Length(Batch) = W_{Length}(x_l \ldots x|Batch|) + Batch_{Length} \tag{8}$$

where avg(.) indicates average pooling within mini-batches, $W_{length} \in RC \times x$ and $Batch_{Length} \in R_x$ are the linear transformation weights and biases for the number of classes C, and 'x' is the dimensionality of input embeddings. Additionally, $x_i$ is a one-hot label vector for the instance $x_i,[;]$. These values are represented by the concatenating function and batch length, respectively. Notably, this work chooses a uniform distribution for the inference phase to produce adapters that are not biased toward dominating classes because the test data labels are unavailable. By adopting a more comprehensive perspective, considering the contextual information in the data can help improve this work understanding of each customer (*e.g.*, languages, text styles). To be more precise, layer-specific adapters are created by extracting contextual information from each layer. By averaging word vectors over the lengths using $l_2$ normalization, context embeddings are extracted, drawing inspiration from the sentence

embeddings. Assuming that the sample vectors $x_j$ from the PLM's $l^{th}$ layer are $f_l(x_j)$, the $l^{th}$ layer's context embeddings can be obtained in Eq. (9):

$$f^l(Batch) = W_f Max(f^l x_1, \ldots, f^l x_{Batch}) + Batch_f \tag{9}$$

where $W_f\ R_d \times x$ and $Batch_f = \in R_x$ are the linear transformation weights and biases, respectively, and $Max(.)$ indicates the max-pooling over the batch. Two types of embeddings are added together to provide comprehensive client embeddings. To further encourage the generator to encode more diversified layer-wise information, the study additionally appends layer-index embeddings into the client embeddings of each layer.

### Client-conditional hyper networks

This research customizes the adapters for each heterogeneous client based on the client embeddings (*Sattler et al., 2020*). This work describes the client-conditional hypernetworks, which produce adapter parameters by taking the client embeddings lab as inputs. They are inspired by the idea of hypernetworks that generate parameters based on supplied input embeddings. Formally, Eq. (10) shows how hypernetworks act to create the adapter parameters (*i.e.,* $U_l$, $D_l$):

$$U^l_{Batch} D^l_{Batch} h(I_B) = (W_U, W_D) I^{Length}_{Batch} \tag{10}$$

where the weights for the hyper networks are represented by the letter 'I', which stands for the input embedding $x$, $W^{length}_D = R^{r*d*s} W^{length}_U = R^{d*r*N}$. It should be noted that various layers exchange hypernetworks with encoded layer-specific data for input embedding.

### Factorization of hyper networks

Although hypernetworks can be used to create customized adapters, hypernetworks usually include many parameters. Thus, the proposed hypernetworks are factorized into two smaller weights (*Fotouhi et al., 2024*). The resulting parameters are also not skewed towards any of the local majority classes in the data distribution of the client because the resultant matrices from the factorized components are $l_2$ normalized. Two factorized parts are used to recreate the up-projection weights in Eq. (11) formally:

$$U^l_{Batch} = WUI_{BBatch} = \partial(f_U s_U) I_{Batch} \tag{11}$$

where x, $\alpha(.)$ indicates the normalization and $f_u = R^{d*s}$, $S_u = R^{d*s*r}$ shows the factorized components from WU with latent factor x.

Regarding factorization, the expressivity and complexity of the resultant adapters are determined mainly by the hidden factor '*s*.' To account for the increased dimensionality of latent components, the two projection weights are coupled in the same way as if they were tied auto-encoders (*i.e.,*) $D^{Length}_{Batch} = U^{Length}_{Batch}$. This method allows the memory needs to be met without sacrificing task precision.

### Aggregation phase

The corresponding trained models are sent back to the centralized server to update the global model when the training step on each client's data is complete. Every client sends the layer-index embeddings and the hypernetwork parameters to the server, updating the global hypernetworks since the training models are hypernetworks.

***Template-based QA (TQA)***

The focus of quality assurance (QA) techniques has historically relied on extracting responses from raw text. Ontologies are used to annotate Web resources and enhance retrieval using query expansion (*Enesi et al., 2019*). Because of this (*Duan et al., 2021a*; *Duan et al., 2021b*), TQA was selected to train a recursive neural network model for template categorization. To obtain the answer(s), a SPARQL query template is essentially the last query's blueprint, which must be created from the question. Query building, relationship extraction, and named entity recognition and disambiguation are most QA systems' primary quality assurance activities. There will only be a solution that works flawlessly in some situations or domains (*Bao et al., 2019*). As a result, specific environments for which the QA components are experts have been developed. These components may be bootstrapped into modular question-answering pipelines. Most quality assurance systems do the conversion of queries into triples and compare them to a current knowledge base using similarity or ranking criteria to obtain the answer. Nevertheless, these triples frequently need to capture the natural language question's semantic structure, leading to incorrect answers or poor SPARQL queries. On the other hand, this work's domain-independent method can be used in any domain with minor modifications to the neural network. The necessary representations are automatically learned by RNN employing labeled instances furnished in the TQA dataset (*Casado et al., 2021*). The dataset uses a list of seed entities and predicates, enabling list filtering to form sub graphs of DBpedia for instantiating SPARQL templates, which in turn provide acceptable SPARQL searches. These SPARQL queries are subsequently used to instantiate Normalized Natural Question Templates (NNQTs) (*d'Hondt et al., 2019*; *D'hondt et al., 2020*), which serve as canonical structures but are often grammatically incorrect. These questions are manually edited and paraphrased by researchers.

# RESULT AND DISCUSSION

## Dataset analysis

A. SQuAD2.0. (Stanford Question Answering Dataset) This work defines the SQuAD Question Answering Task formally as follows: The challenge is to predict the answer span {ts; ts+1;…,te-1; te} given a three-tuple (Q; P; (ts; te)) with question Q, context paragraph C, and start and end indices {ts; te}. Each ai represents an index of the context paragraph corresponding to the answer. For the SQuAD task, a peer-to-peer DL model is created in this article. 10% was used for validation and hyperparameter adjustment, and the remaining 80% of the dataset was used to train the model. To maintain the integrity of the QA models, the last 10% of the dataset is set aside by SQuAD's developers for testing and privacy. This study assessed and thoroughly trained the proposed model on the withheld data using Python as the work's last evaluation tool.

With a few minor encoding stage simplifications, this system also implemented the general structure of the model. Even though the proposed model performed exceptionally well during training, this article finally concluded that, given the time restrictions of this project, the model implementation could have been more practical. This work had around

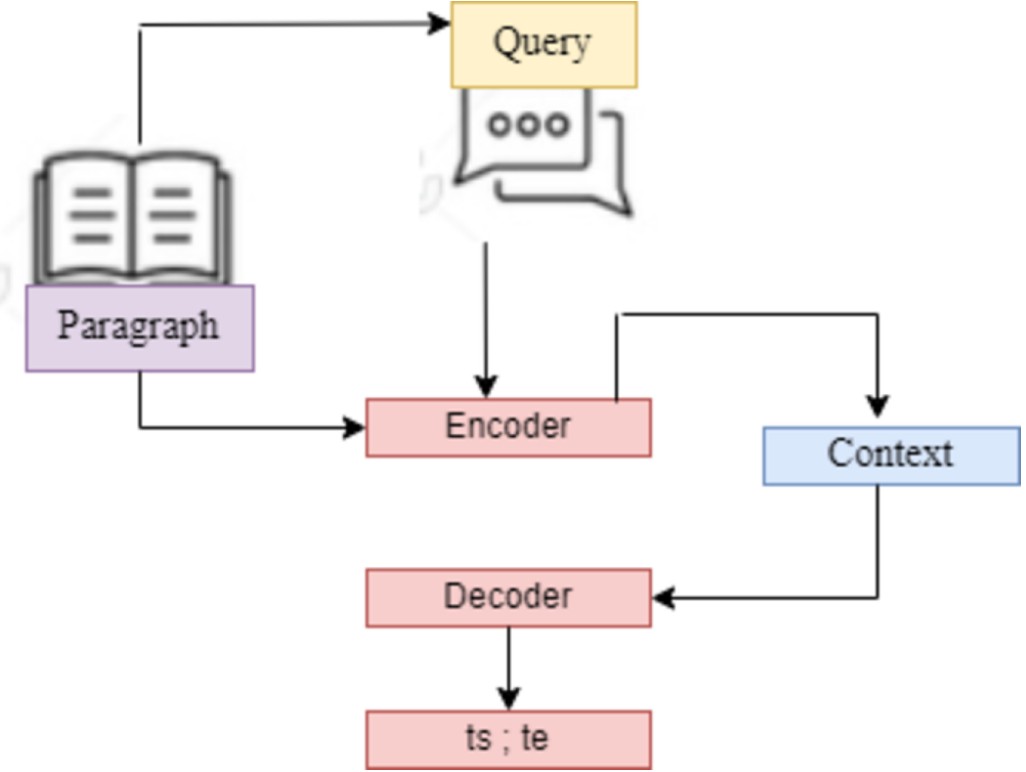

**Figure 4 SQuAD2.0 task based on sequence-to-sequence model.** This models are utilized for tasks that involve the generation of new sentences based on a given input. These tasks include summarization, translation, and generative question answering.

ten days to turn in the project, and this study estimated that for completion, a single epoch under the proposed model BiDAF (*Seo et al., 2016*) algorithm would take about 10 h. Consequently, this article thought that a whole 10-epoch training would take about seven days, which was too expensive. In the following part, this study will review how this model implemented this model and speculate on why it turned out to be too computationally complex for its needs (Fig. 4).

## Model evaluation

Data sets are frequently split into training and test sets to assess ML models. The test data, or held-out samples, are used to evaluate the model and assess its performance objectively. This study assessed each model in each experiment using test data and metric accuracy. Other metrics like precision, recall, and F1-score were not considered except in the trials using heuristics based on F1-scores, as these metrics are better suited for use in situations when there are imbalances in classes. The accuracy metric was suitable since the SQuAD2.0 dataset is balanced.

(a) *Exact Match (EM): This metric measures the percentage of predictions that precisely match any one of the ground truth answers.*

(b) *F1-score: This metric measures the average overlap between the ground truth response and the prediction. By treating the ground truth and the prediction like bags of tokens, this work computes their F1. and gets the average of all the questions by dividing each question's maximum F1 by the total number of ground truth answers.*

The two primary metrics frequently utilized for QA systems are exact match (EM) and F1-score, employed for the evaluation. For each pair (Q, A), these metrics are calculated. Overall, for potential correct answers, the maximum score is calculated if a question has more than one valid answer. If the characters of the model's prediction (EM) match the personalities of (one of) the true answer(s) for each pair (Q, A), then EM = 1; otherwise, EM = 0. This article evaluated the aggregated averaged global model in the centralized trials on the test set after every communication round. This study assessed the FL experiments conducted by peers. The tenth communication cycle is what comes after each client's model. Due to computing expenses, this task was conducted every tenth round, as opposed to simply one global model in the centralized FL experiments, because it required the evaluation of 100 client models on the test data. The peer-to-peer FL experiments included 100 individual models; therefore, it was also necessary to examine how the accuracies of the models varied. As a result, this study looked at the model's accuracy for each of the 100 customers on every tenth communication cycle. This work also computed the model accuracy average in the peer-to-peer experiments. Here is the Eq. (12):

$$Accuracy_{ModelAvg} = \frac{\sum_{x=1}^{x} Accuracy_{modelAvg(W^x)}}{x}. \tag{12}$$

In this case, '$x$' stands for the total number of clients in the network. Using the supplied model weights $w_x$, the function *ModelTestAcc* determines the model test accuracy. In the experiments, assume led, $x = 100$.

Token overlap between the tokens in the correct answer and the expected answer is represented by precision, and recall represents the percentage of tokens in a correct answer, which were accurately predicted in a question. False positive (FP) indicates tokens that do not give the correct answer; however, in the expected answer, true positive (TP) indicates tokens that are identical between the correct answer and the predicted answer, and false negative (FN) displays tokens, which are not in the expected answer but are in the correct answer.

### Communication costs

Following *McMahan et al. (2016)*, this article preserved the communication round number at which a target model accuracy of 92.8% had been attained in every experiment. By using the accuracy of the target model as a benchmark, this study assessed the communication costs across many studies. For each experiment, this work counted the number of models sent in the network at the round, where a 92.8% model accuracy was obtained to determine the communication cost. The number of models provided by the central server and each client was contained in FLAvg and was calculated in Eq. (13):

$$CentralizedFL = N * R * X * 2 + x \tag{13}$$

where $X$ is the total number of clients in the network, $\times$ is the percentage of clients the central server communicates with, and $N$ denotes the round.

When 97% average model accuracy was obtained in the FLAvgP2P experiments, the world tallied all the models submitted by each client, which was computed as follows Eq. (14):

$$FLE2E = R*N*A*X. \tag{14}$$

In this case, $N$ is round, and 92.8% is the percentage of neighbors a client communicates with. Since it is a complete graph, the number of neighbors a client has, represented by the symbol $A$ (equal to 99 in all experiments), is the same for all. Lastly, $X$ represents the total number of network clients.

### Accuracy

The correct answers the FL system gives for a percentage of questions are called Accuracy. There is only one correct response for every question. The span prediction task's accuracy is similar to EM and may be calculated using Eq. (15) as follows:

$$Accuracy = EM = \frac{No\ of\ Correct\ Answer}{No\ of\ Questions}. \tag{15}$$

Recall is the proportion of tokens in a correct answer that have been successfully predicted in a question, whereas precision indicates the percentage of token overlap between the correct answer and the predicted answer.

The TP indicate

s the tokens that are identical between the predicted and correct answers, the tokens that are not in the correct answer but are in the expected response are marked by the FP, and the tokens that are not in the expected answer but are in the correct answer are shown by the FN. Using Eqs. (16) and (17), the precision and recall can be calculated as follows:

$$Precision = \frac{No\ of\ TP}{No\ of\ TP + No\ of\ FP} \tag{16}$$

$$Recall = \frac{No\ of\ TP}{No\ of\ TP + No\ of\ FP}. \tag{17}$$

An indicator of the accuracy of a test is the F1 score. It is the precision and recall weighted average. It provides the calculation for this score. It is calculated in tets instance by comparing each word in the prediction to each in the true answer. The F1-score is based on how many words the prediction and the truth share, Eq. (18)

$$F1 = 2*\frac{Precision + Recall}{Precision + Recall}. \tag{18}$$

The 15,000-question pre-processed dataset was divided into train and test datasets, with 80% of the data being training and 20% being test. There were 13,936 questions in the training dataset and 984 questions in the test dataset. On the test SQuAD and QALD-7

datasets, this RNN model's obtained template classification accuracy was 92.8% and 61.8%, respectively. Equation (19) is used to calculate the accuracy:

$$Acc(x, \overline{x}) = \frac{1}{N} \sum_{i=1}^{N} 1(\overline{x}_i = x_i) \tag{19}$$

where, for example, is the predicted value of the ith and "x" is the matching reality value. There is a total of N samples. Table 1 tabulates the hyperparameters of the model. The input vector was the 444-dimensional word vector concatenated. The optimizer used was the Adam Optimizer, with a small batch size of 25 instances. The loss function employed was cross-entropy loss, which has been shown to work better for tasks involving multivariate classification. Due to the limited training instances and periodic learning rate curtailment, the model needed strict regularization to prevent overfitting and improve its generalization performance. To do this, three strategies were employed:

1. **Weight decay:** The weights are modified by a part of the weight update rule known as Weight Decay, or $l_2$ regularization, following each pass by calculating the product of a factor smaller than 1. This may be considered gradient descent on a quadratic regularization term (comparable to $l_2$ Normalization) and keeps the weights from getting too big. $2.25 \times 10^{-3}$ was the weight decay utilized in the model.

2. **Dropout:** Powerful machine learning systems are deep neural networks that contain numerous parameters. One problem with these linear networks is overfitting. During training, randomly removed units and their connections are included in the neural network's dropout. This mitigates overfitting by keeping the model from forming intricate co-adaptations on the training set. The dropout in the model is 0.2.

3. **Adaptive learning rate:** It was discovered that $1 \times 10^{-2}$ was the ideal initial learning rate for the model. However, after extensive testing in later epochs, it was found that, on the test dataset, when the model's performance peaked, it rapidly overfitted the training dataset. To counteract this, a constant factor periodically decreases the learning rate after a predetermined number of epochs. Once every two epochs, the model was added with a step decay factor of 0.25 to keep the model from overfitting the dataset.

## Model comparison

(1) Compared to individual models like RE2, ESIM, and BERT, the typical federated learning approach FLAvg performs better overall regarding MAP and MRR. The results show that the FL technique, rather than training the model of a single participant on the data, may be able to train a more accurate QA model by using helpful information from several participants. (2) In some domains, individual models may perform better than FLAvg, LG-FLAvg, and FLP2P. For instance, BERT's comparative development over the FLQA set based on MAP is approximately 3.13% compared to FLAvg. One possible explanation is that FLAvg trains a single model for each client, making modeling the FLQA benchmark's statistical heterogeneity challenging. (3) When compared to the original FL frameworks (FLAvg, LG-FLAvg, and FLP2P), CoverQuery's performance has significantly decreased. According to the results, FL offers a higher level of privacy guarantee and is more successful than standard privacy-enhanced approaches. (4) With its proposed *F*LMatch

**Table 1   Model parameter.**

| Parameter | Value |
|---|---|
| Input dimensions | $444 \times 1$ |
| LSTM | $150 \times 1$ |
| Epochs | 7 |
| Mini batch size | 25 |
| Learning rate | $1 \times 10^{-2}$ |
| Weight delay (Regulation) | $2.25 \times 10^{-3}$ |
| Embedded learning rate | $1 \times 10^{-3}$ |
| Dropout | 0.2 |
| Loss function | Cross-entropy loss |
| Optimizer | Adam |
| Learning rate scheduler | Stepwise learning rate decay |
| Step LR step size | Once every two epochs, |
| Step LR decay | 0.25 |

**Table 2   The advantages of suggested FLMatch over BERT are shown by comparing performance measures and t-tests.**

| | QA TYPE-I | | QA TYPE-II | | QA TYPE-III | | QA TYPE-IV | |
|---|---|---|---|---|---|---|---|---|
| **Models** | **MAP** | **MRR** | **MAP** | **MRR** | **MAP** | **MRR** | **MAP** | **MRR** |
| BERT | 0.7109 | 0.71829 | 0.8919 | 0.82617 | 0.8671 | 0.8719 | 0.8181 | 0.8919 |
| Proposed *F* LMatch model | 0.7516 | 0.7618 | 0.9102 | 0.8178 | 0.8919 | 0.9192 | 0.8367 | 0.9181 |

model, the best performance is attained. To gain general knowledge from multiple clients, the recommended method of breaking down the QA model into a private patch for capturing a shared backbone and the local data characteristics has been validated by the findings.

This work created multiple public QA collections based on the new benchmark dataset FLQA to make FL for QA research easier (Table 2).

- QA TYPE-I is a corpus of information regarding mobile applications' privacy policies that span many categories. Crowd workers inquire about a particular mobile application's privacy. The writers then assemble seven professionals with legal backgrounds to craft answers to questions.
- QA TYPE-II is a biomedical semantic indexing and QA competition. Biomedical professionals can voice their information demands and will receive succinct responses that synthesize data from many sources.
- QA TYPE-III For the financial opinion mining and quality assurance task of WWW'18. Instead of economic data, the model uses Task 2 data—Opinion-based QA. The questions are answered based on a corpus of documents from multiple financial data sources.
- QA TYPE-IV, motivated by the strong commercial and scientific interest in the insurance domain, QA pairs in this area. Professionals possessing extensive topic knowledge have produced responses to the questions gathered from real-world users.

## Performance analysis

This work achieved 75.18% F1 and 56.19% EM scores, submitting the best model to the test set leaderboard. The best model was a combination of an encoder that used co-attention and a decoder that used CNN, the highway network, and a search mechanism. For brevity, this work refers to this best model as UMHFLM.

### Performance across length

Figure 5 displays the F1-scores and the exact match (EM) for answers up to 15 tokens so that you can assess how well the algorithm performed while predicting responses of varying durations. This proposed model anticipates that the model's performance will deteriorate with increasing answer length, as is the case with many NLP applications, including neural machine translation. Nevertheless, for answers up to length 12, there is no discernible decline in the F1-score. However, the model observes a steady decrease in accuracy for both EM after that. This makes sense intuitively since the number of words increases, and the computation of the proper answer span gets harder. The graph shows that the EM and F1 scores decrease as response lengths do. Furthermore, this work notes that when the answer length increases, the difference between F1, EM, and accuracy drops at distinct rates.

## Performance across question head

This article also looks at how well this model performs with various question head terms. Figure 6 shows that queries that begin with frequently asked words, including *{'HOW'; 'WHAT'; 'WHEN'* } have high F1-scores. However, this research also notes that the model could do better when asked questions that begin with the terms {*'WHY'; 'WITH'* }. This work also observes a significant distinction for {*'WHY'* } questions between F1 and EM. Finding the answer's critical portion is simpler for questions of this type, but determining the precise answer's range is considerably more difficult. Interestingly, other published *State-Of-The-Art* models also exhibit subpar performance on "*WHY*" questions, suggesting that obtaining high accuracy for this type of question is inherently challenging.

### Performance across questions with non-interrogative head

This work could not locate any research that looked at the system's performance for questions that started with non-interrogative words, even though it is usual for current literature to analyze performance across questions beginning with different interrogative words. These questions are typically more sophisticated and rhetorical and are called non-interrogative. This article examined how well the proposed model performed for these questions, and Fig. 7 displays the results for several of them. This work finds that this model's effectiveness varies significantly across different start words, but this study can identify some intriguing events. Figs. 8 and 9 illustrates how effectively this model works for questions that begin with "about" and "*approximately.*" The stark contrast between the F1-score and EM for questions that start with "although" is the most intriguing finding. This work may easily anticipate that questions beginning with "*although*" will be structurally complicated sentences based on how this research uses the language daily. Based on the performance of this model on questions of this type, the model can observe that while it is

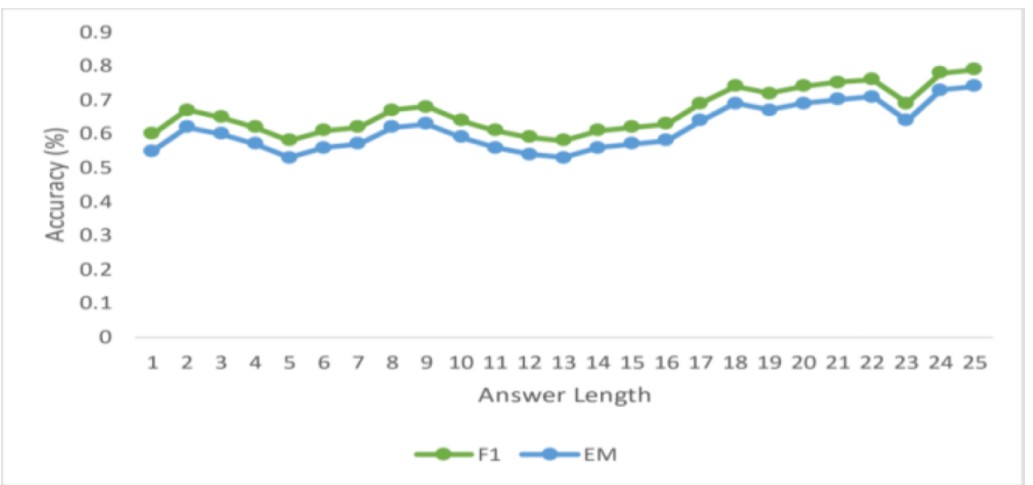

**Figure 5** **Variations in the ground truth answer length (up to 15) for CCHNS.**

challenging to obtain the precise answer in this instance, this model performs remarkably well in getting the general idea of the solution. This work believes that further research on enhancing performance for non-interrogative questions will result in systems that can comprehend natural language's complex meaning. Ultimately, the model obtained the following outcomes: "*Test*" denotes the hidden test set kept up to date by '*y*', the SQuAD2.0 creators.

### F1 and EM scores

Unfortunately, the performance of this BiDAF model was not up to pace. The final F1 and EM scores for the training set were approximately 10% and 6%, respectively. For EM and F1, the approximate final validation scores were 3.7% and 7.6%, respectively. The outcomes outperform this model's initial baseline. Plots of the data collected during the four training epochs are shown above. According to this study, the model's average loss and global norm decrease slowly while its F1 and EM scores rise shown in Table 3. Given the low learning rate in these early epochs, the proposed model F1 and EM scores may move slowly. The model in this work was trained using a fixed learning rate of 0.0001. This article should have started with a more significant learning rate, especially during the first epoch when the model does not have to care about fine-grained learning, even though a low learning rate is best for later epochs. Although tinkering with the initial value and decay rate would have taken a lot of time, having an exponentially declining learning rate would have been nice. This model's sluggish development was also caused by the massive number of parameters it had to learn.

This is why the research developed word vectors that effectively capture syntactic and semantic information using trainable 300d GloVe vectors. It sounds okay, but the 41 m parameters the model had to learn were significantly increased. Due to time constraints, it is only possible to train for four epochs, which took roughly eleven hours each. This model was learning appropriately based on the global norm and diminishing loss of this

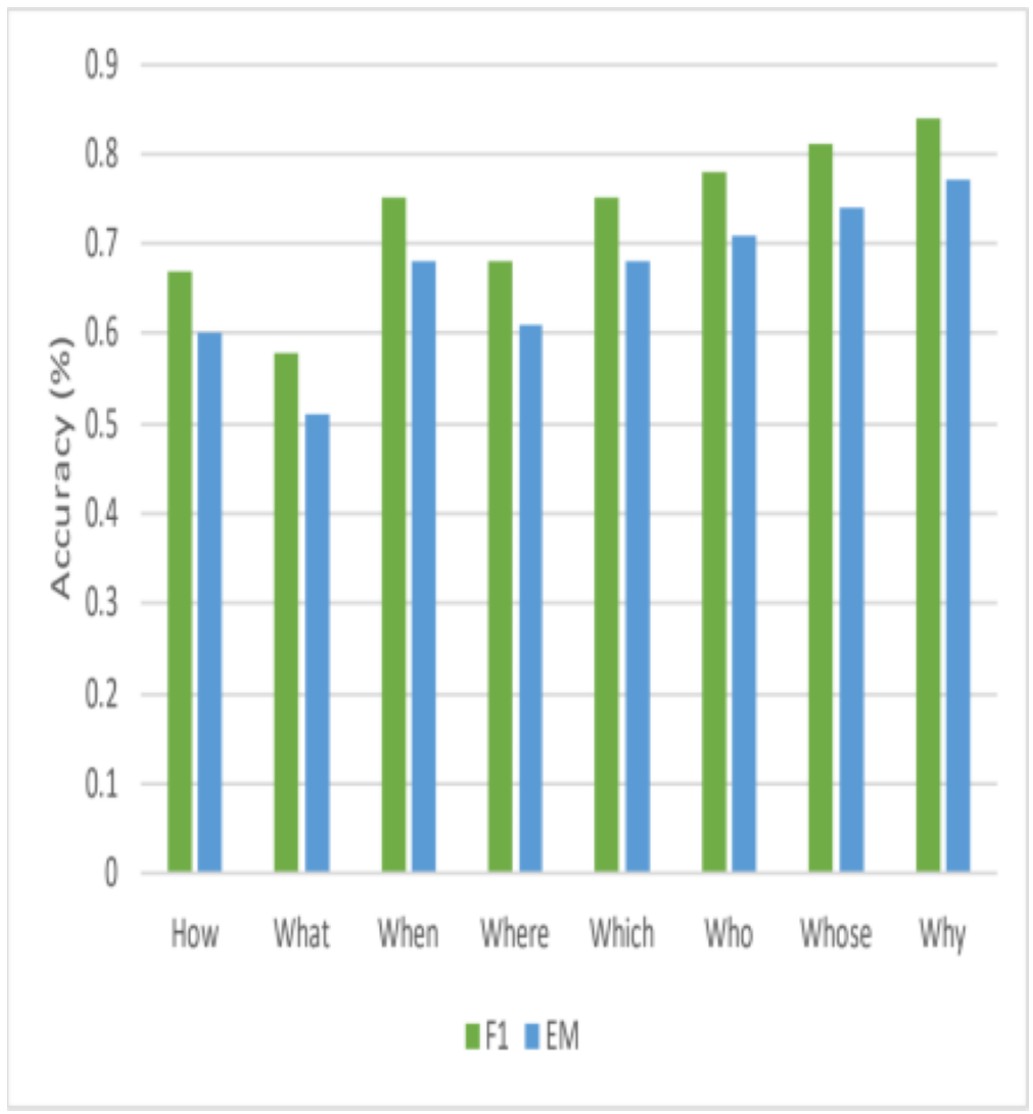

**Figure 6** Comparisons of performance for various question heads.

work, in addition to its F1/EM scores. This work would expect these measurements to be relatively low once the model has completed training and the optimal set of parameters has been identified. Nevertheless, considering the concave F1 and EM curves, the model predicts that the scores would not increase significantly. This can point to a severe issue with the model or a flaw in the test code. For a reasonable model to overfit the training set and obtain noticeably greater training accuracy, tens of millions of parameters should be plenty.

### Time performance

The most significant challenge to testing, refining, and training the models proposed in this study was time. Testing minor or hyperparameter adjustments was particularly challenging

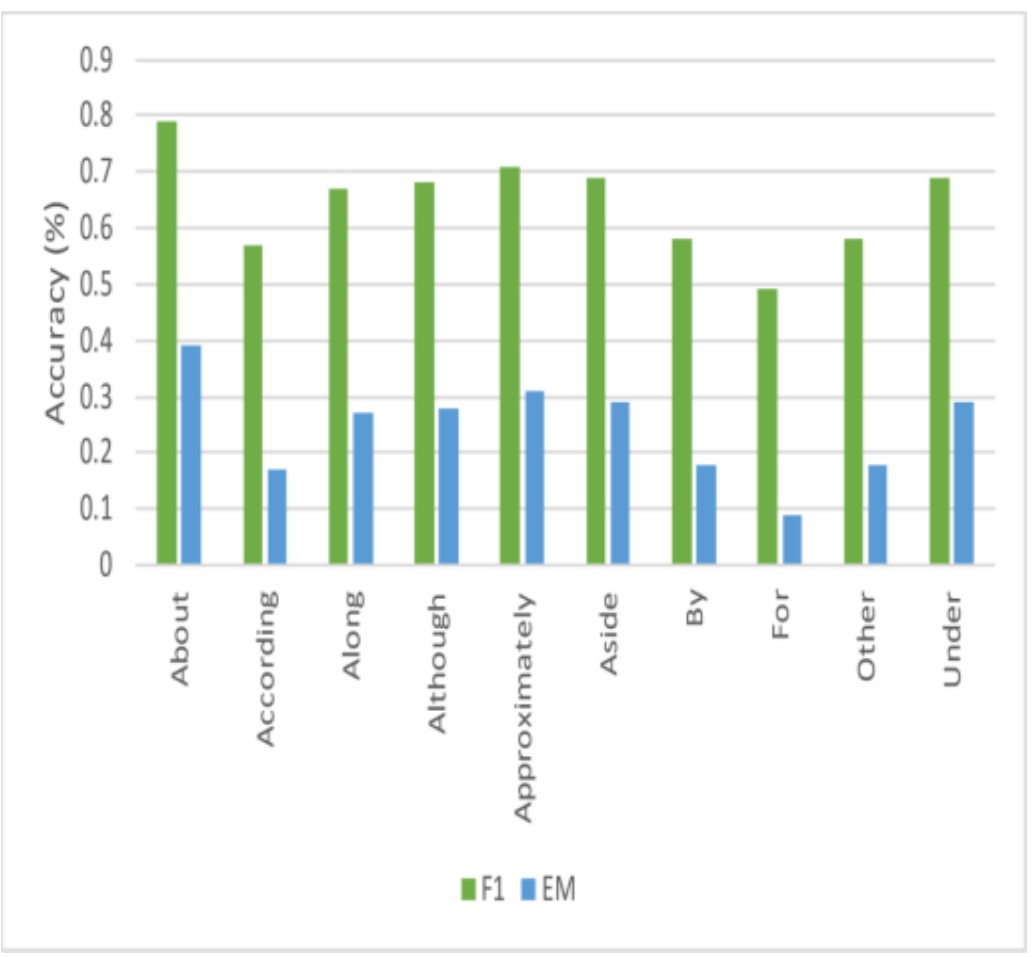

**Figure 7** Comparisons of performance for questions that are non-interrogative.

because the recommended BiDAF model required more than 10.5 h per GPU epoch to train. They asserted that forgetting errors constituted the primary source of this research's limitations. A batch size more significant than four resulted in out-of-memory issues that may manifest hours into training on the forty million parameter model utilized in this work. Looking back, this study should have run a much larger batch using smaller GloVe vectors or restricting the context paragraph size to a few hundred. An evaluation of the model's bidirectional dynamic RNNs with the setting "Swap Memory = True" revealed no discernible effect on memory efficiency. The model also carefully examined test code to hunt for explicit for-loops that might be utilizing excessive amounts of memory or carrying out tensor calculations inefficiently; nevertheless, nothing unusual was found. The attention flow layer, which is the core of the BiDAF model, also significantly prolonged the training duration. This layer consumed many processing resources since it created the attention layer's similarity matrix from the 200-dimensional hidden states of every word in the question and context paragraph.

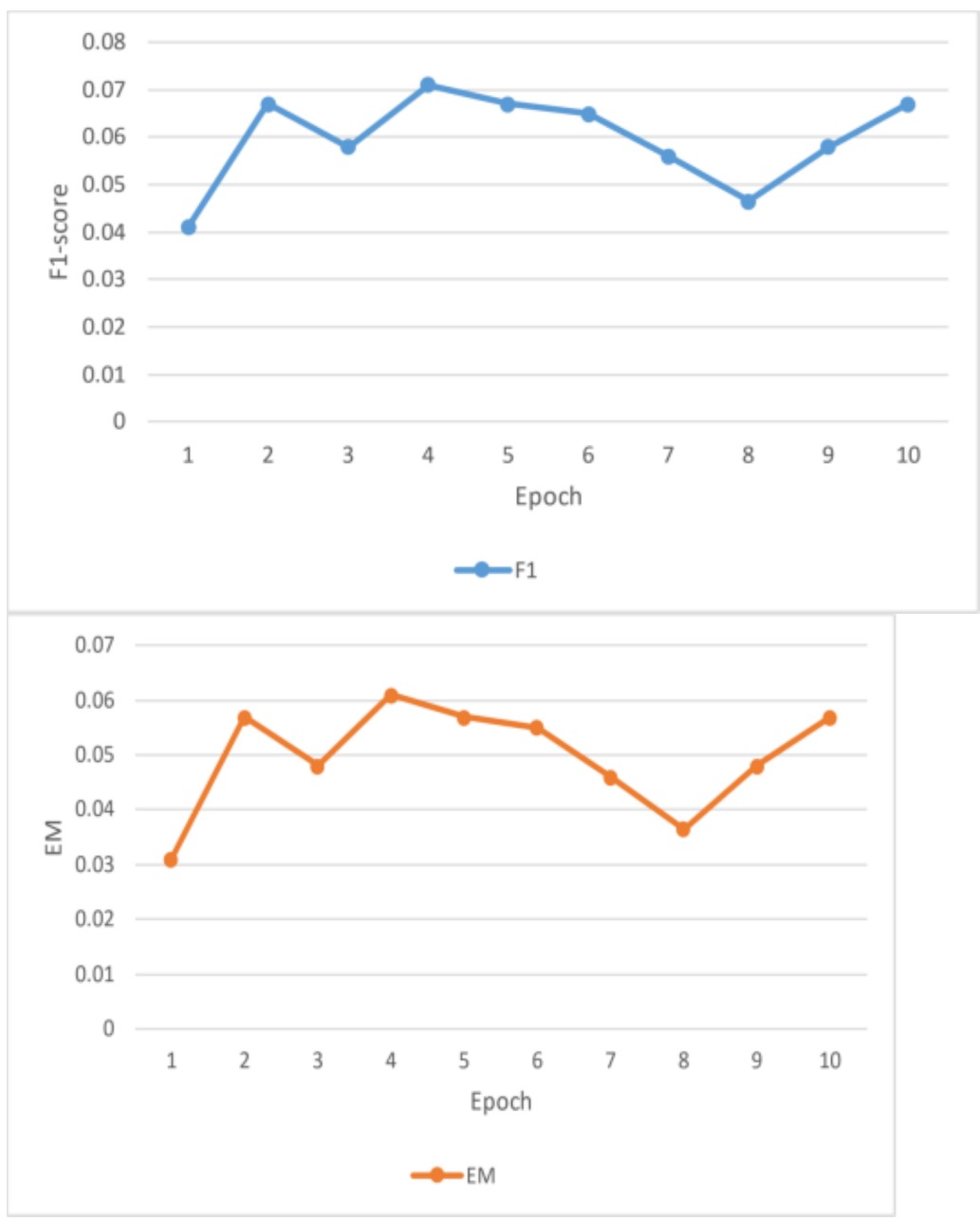

**Figure 8** **Result analysis F1 and EM(a).**

The used SQuAD2.0 (Stanford Question Answering Dataset) benchmark dataset for evaluating machine comprehension and question answering systems. Unlike its predecessor, SQuAD1.1, SQuAD2.0 incorporates unanswerable questions, adding a layer of complexity that better reflects real-world scenarios. This enhancement encourages models to not only identify answers within a given context but also recognize when questions cannot be answered, fostering more robust and accurate performance evaluation. By including

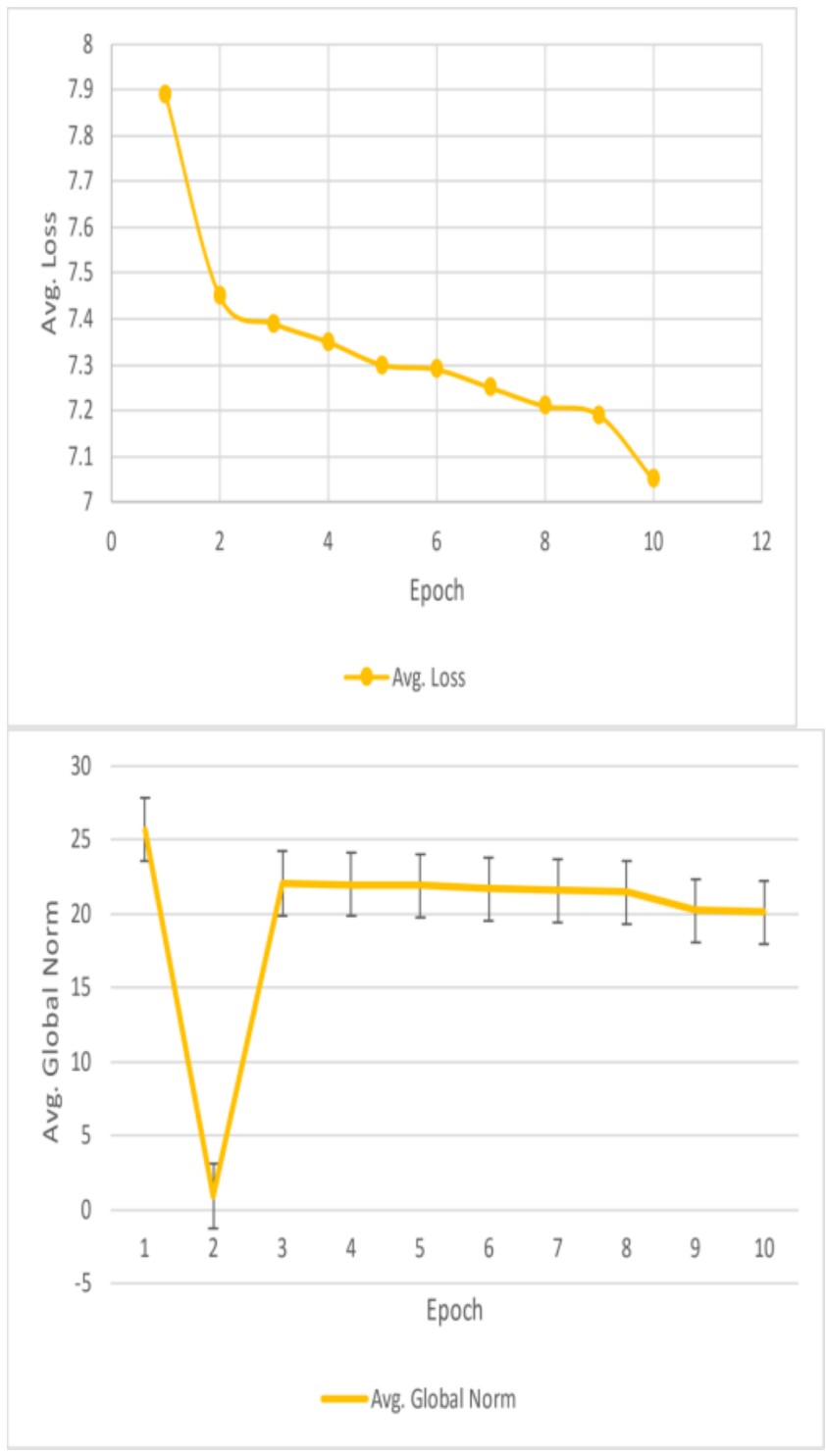

**Figure 9** **Result analysis losses and global Norm(a).**

**Table 3  Analysis of F1 and EM scores.**

| Performance measures | Training | Validation | Dev | Test |
| --- | --- | --- | --- | --- |
| F1 | 76.25% | 59.78% | 54.19% | 55.91% |
| EM | 62.23% | 44.18% | 40.4% | 41.4% |

unanswerable questions, SQuAD2.0 promotes the development of AI systems capable of discerning between answerable and unanswerable queries, thus advancing the state-of-the-art in natural language understanding and question answering. In future, we plan to validate the performance of the proposed model on large scale real time dataset.

# CONCLUSION AND FUTURE WORK

QA learning is enhanced by the Federated Learning Matching framework $F$LMatch, which converts models into shared and private modules utilizing local data and shared knowledge. This machine learning technique uses collaborative training, private incremental learning, and a private data pipeline to address data-sharing concerns. By maintaining privacy-preserving information, the $F$LMatcQA benchmark dataset enhances QA performance by simulating many real-world circumstances. By considering client data distribution, the proposed hyper network-based FL framework, the User Modified Hierarchical Federated Learning Model (UMHFLM), reduces the number of communication rounds between edge nodes and centralized servers by generating client-customized adapters. The $F$LMatch FL method offers semantics and uniformity for superior classification representations to manage data heterogeneity and class imbalance. The work uses recursive neural networks (RNN) to convert QA tasks over linked data into template classification and slot-filling tasks. For the SQuAD2.0-QA challenge, a Deep Learning model demonstrated encouraging initial results but had trouble responding to interrogative and non-interrogative questions. According to the research, neural models lack information on the learning process, making debugging difficult. Even well-designed models might take hours or days to train, which can result in out-of-memory catastrophes. The RNN model finds 38 templates using SQuAD2.0 datasets, and it classifies the two most likely templates with 92.8% and 61.8% accuracy, respectively. With a MAP margin of 2.60% and an MRR margin of 7.23% at 100% and 20% data ratios, $F$LMatch surpassed $B\,E\,R\,T$ in a trial. This work submitted the best model to the test set leader board with 75.18% F1 and 56.19% EM scores. Hyper-parameter tweaking, training intricate ensemble models, investigating CNNs for character-level embeddings and attention, and using reinforcement learning concepts from other AI domains can all be used to enhance the SQuAD2.0 challenge in natural language processing. One method is to use grammar and syntactic structure to design rules that resemble Markov decision processes.

## ACKNOWLEDGEMENTS

The authors acknowledge their universities for providing access to research facilities and academic consultation.

### Funding

The authors received no funding for this work.

### Competing Interests

The authors declare there are no competing interests.

### Author Contributions

- Saranya M conceived and designed the experiments, performed the experiments, analyzed the data, performed the computation work, prepared figures and/or tables, authored or reviewed drafts of the article, and approved the final draft.
- Amutha B conceived and designed the experiments, performed the experiments, analyzed the data, performed the computation work, prepared figures and/or tables, authored or reviewed drafts of the article, and approved the final draft.

### Data Availability

The data is available in the Supplemental Files.

The third party datasets are available at:

- https://github.com/ag-sc/QALD/blob/master/7/data/qald-7-train-multilingual.json
- https://www.kaggle.com/datasets/thedevastator/unlock-smarter-querying-with-lc-quad-2-0
- https://huggingface.co/datasets/rajpurkar/squad
- https://www.kaggle.com/datasets/parthplc/squad-20-csv-file

### Supplemental Information

Supplemental information for this article can be found online at http://dx.doi.org/10.7717/peerj-cs.2092#supplemental-information.

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
