# Peer review of "FLMatchQA: a recursive neural network-based question answering with customized federated learning model"

_PeerJ Computer Science, doi:10.7717/peerj-cs.2092_

## Round 0.1 · original submission · Major Revisions

Based on the reviewers' comments, the manuscript must be revised.

**Language Note:** The review process has identified that the English language must be improved. PeerJ can provide language editing services - please contact us at [email protected] for pricing (be sure to provide your manuscript number and title). Alternatively, you should make your own arrangements to improve the language quality and provide details in your response letter. – PeerJ Staff

Reviewer 1 ·

Basic reporting

The manuscript entitled “FLMatchQA: A recursive neural network based question answering with customized federated learning model” has been investigated in detail. The paper proposes an approach integrating Federated Learning (FL) and Question Answering (QA) systems to address privacy concerns in sophisticated data access. It introduces a User Modified Hierarchical Federated Learning Model (UMHFLM) that selects local models for user tasks, employing Recurrent Neural Networks (RNN) for automatic learning and question categorization. Evaluation on the SQuAD-2.0 dataset demonstrates the model's performance, achieving a template classification accuracy of 92.8% and 61.8% on the SQuAD2.0 and QALD-7 datasets, respectively. FLMatch outperforms BERT significantly, with a Margin of Average Precision (MAP) of 2.60% and a Margin of Relative Ranking (MRR) of 7.23% at 20% data ratio. There are some points that need further clarification and improvement:
1) The problem statement lacks clarity regarding the specific challenges addressed by the proposed research. The introduction should clearly outline the limitations of existing AI techniques in QA systems and the rationale for exploring Federated Learning (FL) as an alternative.
2) The methodology section requires more detailed explanations of the hierarchical FL systems and the development of client-specific adapters. There is a lack of clarity on how the User Modified Hierarchical Federated Learning Model (UMHFLM) operates and selects local models for user tasks.

Experimental design

The technical details regarding the utilization of RNN for automatic learning and question categorization are insufficiently explained. Readers need a more comprehensive understanding of how RNN models are trained and applied in the proposed framework.

While the paper mentions the evaluation of the model using the SQuAD-2.0 dataset, it lacks clarity on the specific evaluation metrics used and how the performance of the proposed model compares to existing methods.

Validity of the findings

The evaluation of the proposed model is limited and lacks comprehensive validation against existing benchmarks and datasets.

The statistical analysis provided to support the model's performance is insufficient, and the significance of the results is not adequately demonstrated.

“Result and Discussion” section should be edited in a more highlighting, argumentative way. The author should analysis the reason why the tested results is achieved.

Additional comments

The manuscript entitled “FLMatchQA: A recursive neural network based question answering with customized federated learning model” has been investigated in detail. The paper proposes an approach integrating Federated Learning (FL) and Question Answering (QA) systems to address privacy concerns in sophisticated data access. It introduces a User Modified Hierarchical Federated Learning Model (UMHFLM) that selects local models for user tasks, employing Recurrent Neural Networks (RNN) for automatic learning and question categorization. Evaluation on the SQuAD-2.0 dataset demonstrates the model's performance, achieving a template classification accuracy of 92.8% and 61.8% on the SQuAD2.0 and QALD-7 datasets, respectively. FLMatch outperforms BERT significantly, with a Margin of Average Precision (MAP) of 2.60% and a Margin of Relative Ranking (MRR) of 7.23% at 20% data ratio. There are some points that need further clarification and improvement:
1) The problem statement lacks clarity regarding the specific challenges addressed by the proposed research. The introduction should clearly outline the limitations of existing AI techniques in QA systems and the rationale for exploring Federated Learning (FL) as an alternative.
2) The methodology section requires more detailed explanations of the hierarchical FL systems and the development of client-specific adapters. There is a lack of clarity on how the User Modified Hierarchical Federated Learning Model (UMHFLM) operates and selects local models for user tasks.
3) The technical details regarding the utilization of RNN for automatic learning and question categorization are insufficiently explained. Readers need a more comprehensive understanding of how RNN models are trained and applied in the proposed framework.
4) While the paper mentions the evaluation of the model using the SQuAD-2.0 dataset, it lacks clarity on the specific evaluation metrics used and how the performance of the proposed model compares to existing methods.
5) The language and terminology used throughout the paper are confusing and often convoluted, making it difficult to understand the proposed approach clearly.
6) There are several instances of typographical errors and unclear phrasing, which detract from the readability of the paper.
7) The paper lacks a clear structure, making it challenging for readers to follow the flow of ideas and understand the methodology proposed.
8) The transition between sections is abrupt, and the logical progression of the argument is unclear.
9) The paper lacks detailed explanations of key concepts and methodologies, particularly regarding the implementation of the User Modified Hierarchical Federated Learning Model (UMHFLM).
10) The methodology section is vague and lacks sufficient detail to understand how the proposed approach is implemented and evaluated.
11) The evaluation of the proposed model is limited and lacks comprehensive validation against existing benchmarks and datasets.
12) The statistical analysis provided to support the model's performance is insufficient, and the significance of the results is not adequately demonstrated.
13) “Result and Discussion” section should be edited in a more highlighting, argumentative way. The author should analysis the reason why the tested results is achieved.
14) Figures 5-11 should be improved.
15) It will be helpful to the readers if some discussions about insight of the main results are added as Remarks.
This study may be proposed for publication if it is addressed in the specified problems.

Reviewer 2 ·

Basic reporting

1. Keywords are missing after Abstract

2, At the end of introduction section. Paper sections should be described as given below.

"This paper comprises of five (5) sections. The first section covers the introduction of this research study. Related work with respect to research has been explained in Section 2...."

3. Literature Review section is very very short. Authors have to summarize at least 10 to 12 related articles.

4. The problem statement and Research Objectives should be provided precisely.

5. It is difficult to see the novelty of the work with respect to available literature.

6. Avoid using words "We" and "Our" in the manuscript.

7. The punctuation and grammar of the manuscript should be improved considerably.

8. References are missing in the manuscript

Experimental design

Result and discussion section is badly written. please re-write. Results are not properly visible. No data analysis is there.

Validity of the findings

no comment

Additional comments

Generally speaking, the paper needs more improvement to make it structured and organized so that the following major corrections are suggested.

---

## Round 0.2 · accepted · Accept

The authors addressed all comments accurately, and the manuscript can be accepted.

Reviewer 1 ·

Basic reporting

My comments have been addressed. It is acceptable in the present form.

Experimental design

My comments have been addressed. It is acceptable in the present form.

Validity of the findings

My comments have been addressed. It is acceptable in the present form.

Additional comments

My comments have been addressed. It is acceptable in the present form.